# Degradation and Pathways of Carvone in Soil and Water

**DOI:** 10.3390/molecules27082415

**Published:** 2022-04-08

**Authors:** Chenyu Huang, Wenwen Zhou, Chuanfei Bian, Long Wang, Yuqi Li, Baotong Li

**Affiliations:** 1College of Land Resources and Environment, Jiangxi Agricultural University, Nanchang 330045, China; chenyuhuang999@163.com (C.H.); bcf940331@163.com (C.B.); wl2283807483@163.com (L.W.); 2College of Food Sciences, Jiangxi Agricultural University, Nanchang 330045, China; fly_zww@163.com; 3College of Engineering, Jiangxi Agricultural University, Nanchang 330045, China; tonglm66@163.com

**Keywords:** carvone, soil degradation, photolysis, degradation products

## Abstract

Carvone is a monoterpene compound that has been widely used as a pesticide for more than 10 years. However, little is known regarding the fate of carvone, or its degradation products, in the environment. We used GC-MS (gas chromatography–mass spectrometry) to study the fate of carvone and its degradation and photolysis products under different soil and light conditions. We identified and quantified three degradation products of carvone in soil and water samples: dihydrocarvone, dihydrocarveol, and carvone camphor. In soil, dihydrocarveol was produced at very low levels (≤0.067 mg/kg), while dihydrocarvone was produced at much higher levels (≤2.07 mg/kg). In water exposed to differing light conditions, carvone was degraded to carvone camphor. The photolysis rate of carvone camphor under a mercury lamp was faster, but its persistence was lower than under a xenon lamp. The results of this study provide fundamental data to better understand the fate and degradation of carvone and its metabolites in the environment.

## 1. Introduction

The use of pesticides has become an essential component of agricultural production due to their proven efficacy in improving crop yields and quality, thus enabling producers to meet an ever-rising demand for food [1]. However, off-target toxicity and environmental pollution caused by the mis- and overuse of pesticides has rightfully begun to raise concern [2]. In an answer to these concerns, environmentally-persistent pesticides have been gradually replaced by environmentally-degradable pesticides. When applied, pesticides enter the soil, water, and atmosphere, where a combination of intrinsic and environmental factors determines their fate. Pesticides can be retained within the immediate area, migrate to different areas, or be transformed and degraded into other compounds [3]. Degradation, whether through biotic or abiotic processes, generally results in the breakdown of pesticides into non-toxic or less-toxic compounds [4].

Degradable pesticides are presumed harmless; however, their residues and degradation products may still have harmful effects on non-target organisms and the environment [5]. For example, the photodegradation products of alloxydim [6,7], a post-emergence herbicide, show a higher toxicity than the parent compound, and the degradation products of benzothiazole (TCMTB), a broad-spectrum biocide, show high toxicity to aquatic organisms. Therefore, in order to fully understand the fate of pesticides in the environment and the risks they may pose, it is necessary to study not only the parent compound but also the degradation products of pesticides in different environments.

The potential use of monoterpenoids, found in plant essential oils, as pesticides has been studied for many years. Compared with traditional synthetic pesticides, monoterpenoids have the advantages of abundant raw materials, limited insecticide resistance, and environmental friendliness [8]. Monoterpenoid pesticides tend to have chemical and physical attributes which make them highly environmentally degradable compared to traditional synthetic pesticides [9,10]. Specifically, monoterpenoid pesticide compounds tend to have a low molecular mass and a cyclic structure and also tend to be aromatic and volatile at near-room temperature. These attributes make monoterpenoid pesticides prone to oxidation, cyclization, isomerization, dehydrogenation, and other decomposition reactions [10], rendering them environmentally unstable.

Monoterpenoids emitted into the atmosphere tend to form aerosols, inducing photochemical degradation [11]. For example, the monoterpenoid limonene is oxidized into carvone, limonene oxide, carveol, and limonene hydroperoxide in the atmosphere [12]. In soil and water, monoterpenoids are primarily degraded by microorganisms, while a minority may be lost to leaching and evaporation [13]. This degradation process will be affected by environmental conditions, such as soil type, pH, temperature, humidity, precipitation, etc. [14]. Previous research on the degradation of monoterpenoids produced by *Myrtis communis* in the soil found that decomposition accelerated during periods of high microbial activity [15]. Moreover, some monoterpenoids can achieve the bioremediation of contaminated soil by promoting the degradation of organic pollutants through increasing microbial activity and residency time [16].

The monoterpenoid carvone has shown potential as an insecticide, fungicide, antioxidant, and plant growth regulator [17,18,19,20]. Carvone naturally occurs in parsley oil (S-isomer), dill seed oil (S-isomer), and spearmint oil (R-isomer) [21]. Carvone exists as two isomers, d-carvone and l-carvone, and the enantiomeric nature of the compound has been fully analyzed [22]. Previous research has found that carvone has low persistence in soil [23]. Additionally, carvone is converted to carveol in both rat and human liver microsomes [24]. In an ethanol–water solution, carvone has been shown to photodegrade to carvone camphor [25]. The structures of carvone and its degradation productions can be seen in Figure 1.

Because carvone has potential as a broad-spectrum pesticide, it is very important to understand its environmental behaviour and fate in agricultural soils and related environments. We studied the degradation behaviour of carvone in various soil types and light conditions in order to quantify the carvone residues and degradation products. These data can provide reference values for the environmental impact and commercial use of carvone.

## 2. Materials and Methods

### 2.1. Chemicals and Instruments

The following chemicals were used in our experiments: Carvone (purity > 99.3%) was purchased from McLin Co., Ltd. (Shanghai, China). Dihydrocarveol (purity > 99.5%), dihydrocarvone (purity > 99.1%), and carvone camphor (purity > 98.7%) were purchased from Merck Chemical Technology Co., Ltd. (Shanghai, China). Chromatographic grade ethanol and n-hexane, and analytical grade sodium chloride (NaCl), anhydrous magnesium sulfate (MgSO_4_), potassium dihydrogen phosphate (KH_2_PO_4_), sodium hydroxide (NaOH), hydrogen peroxide (H_2_O_2_), and hydrochloric acid (HCl), were supplied by McLin Co., Ltd. (Nanchang, China). Octadecylsilane (C18, 60 μm), PSA (50 μm), and GCB (50 μm) adsorbents were purchased from Bona Air Group Ltd. (Tianjin, China).

The following instruments were used in our experiments: gas chromatography (7890b)-mass spectrometry (5977B) equipped with HP-5MS capillary column (60 m length, 0.25 mm inner diameter, 0.25 μm film thickness, 5% phenyl-methyl polysiloxane) (Agilent Technologies Co., Ltd., Beijing, China); Milli-Q Advantage AW deionized water system (Merck Chemical Technology Co., Ltd., Shanghai, China); OGX-450C intelligent light incubator (Kunning Instrument Co., Ltd., Shanghai, China); EX224ZH electronic analytical balance (OHAUS Instrument Co., Ltd., Shanghai, China); Eppendorf centrifuge 5804R high speed and large capacity refrigerated centrifuge (Eppendorf AG, Hambery, Germany); vacuum dryer (Huixin Chemical Glass Instrument Co., Ltd., Hengshui, China); LDZX-50KBS vertical autoclave steam sterilizer (Shenan Technology Co., Ltd., Shanghai, China); CME-PC photochemical reactor (CME Technology Co., Ltd., Beijing, China); and KQ2200E ultrasonic cleaner (Kunshan Ultrasonic Instruments Co., Ltd., Kunshan, China).

### 2.2. Preparation of Stock Standard Solutions

To make n-hexane standard solution, samples (100 ± 0.1 mg) of carvone, dihydrocarveol, dihydrocarvone, and carvone camphor standards were weighed into 100-mL brown volumetric flasks using an electronic analytical balance and were dissolved in n-hexane. The samples were then subjected to 5 min of ultrasonic treatment with an ultrasonic cleaning machine. After cooling to room temperature, the samples were mixed in equal volumes in n-hexane to obtain a 1000 mg L^−1^ mixed standard stock solution. The preparation of ethanol standard solution of carvone is the same as the above operation. The matrix standard working solutions (0.05–5 mg L^−1^) were prepared by diluting the reserve solutions with blank matrix extract. All matrix standard working solutions and standard working solutions were stored in 10-mL brown volumetric flasks at −18 °C. Each solution was analyzed by GC-MS to ensure that the error value did not exceed the experimental standard.

### 2.3. Collection and Characterization of Soil Samples

Four types of soils were collected from different regions of China: Hubei (S1), Jilin (S2), Sichuan (S3), and Zhejiang (S4). All soil samples were collected from a depth of 0–20 cm, dried naturally, and filtered through a 2 mm sieve. According to the FAO soil classification standards [26,27], these soils were identified as Alisols (S1), Phaeozems (S2), Gleysols (S4), and Anthrosols (S4). The physical and chemical properties of soil samples, as determined according to the method of the Southern United States Cooperative Extension [28], are shown in Table 1.

### 2.4. Degradation of Carvone in Different Soil Types

The abiotic and biotic degradation of carvone in soil was analyzed as follows:(1)Aerobic treatment: 50 ± 0.1 g of soil sample (S1–S4) was weighed in a 250 mL conical flask; 250 μL of 1000 mg L^−1^ carvone ethanol solution was added into it and mixed evenly for 5 min to form an initial carvone concentration level of 5 mg kg^−1^. The soil sample was then covered with a sterile cotton plug, which was opened weekly to maintain the aerobic state of the soil.(2)Anaerobic treatment: Carvone was added according to the above description, and enough sterilized deionized water was added to the soil sample (S1–S4) to form a water layer of at least 2 cm on the soil surface. The soil sample was then covered with a sterile cotton plug and vacuum extracted in the conical flask to form anaerobic conditions.(3)Sterilized treatment: 50 ± 0.1 g of soil sample (S1–S4) was weighed in a 250 mL conical flask, plugged with a cotton stopper, and placed in a high-pressure sterilization pot at 121 °C and 200 kPa for 30 min. After the soil samples were cooled to room temperature, soil respiration was measured by alkali absorption method (AA-method) [29] to ensure complete sterilization. This procedure was repeated until no soil respiration was detected. After this, carvone was added according to the above description.

Incubation was carried out according to the following: 10% (*w*/*w*) deionized water was added to each sample, and the mixture was incubated in a constant temperature and humidity incubator at 25 ± 1 °C for 14 days in the dark. The soil moisture was regulated every two days.

#### Degradation of Carvone in Different Soil Conditions

The degradation of carvone under various soil conditions was analyzed as follows, using S2 soil under aerobic condition:(1)pH: We used 0.1 M hydrochloric acid (HCl) aqueous solution to adjust the pH of the soil to pH 3, 4, and 5. At the same time, a control group without HCl and only deionized water was created.(2)Organic matter: In order to remove the organic matter, 100 g S2 soil was added to a 2 L beaker, mixed with deionized water, and stirred evenly. Then, 30% hydrogen peroxide was slowly added and vigorously stirred to prevent the bubbles from overflowing. After the foam subsided, the beaker was heated in a water bath at 70~80 °C, in order to speed and complete the reaction. This process was repeated until the soil no longer produced foam upon the addition of hydrogen peroxide. Then, 400 mL deionized water was added to the beaker and the mixture was boiled to remove all hydrogen peroxide. This process was repeated until no hydrogen peroxide remained. Finally, soil was dried in a 105 ℃ oven and screened to obtain organic-matter-free S2 soil; 50 g of treated soil was sterilized in an autoclave at 121 °C and 200 kPa for 30 min to ensure both sterilization and complete removal of organic material.(3)Temperature: S2 soil was cultured in incubators at 10 °C, 25 °C, 35 °C, and 50 °C, of which 25 °C was the control condition.(4)Moisture: The water content of S2 was set to 0.1, 10, 20, and 30% (*w*/*w*) using deionized water, of which 25% level is the maximum water holding capacity of the soil.

### 2.5. Photolysis of Carvone

The light-induced degradation of carvone was analyzed as follows:

In order to create a buffered solution for photolysis testing, 500 mL 0.1 mol L^−1^ potassium dihydrogen phosphate solution was weighed in a 1000 mL beaker, treated with 296.3 mL of 0.1 mol L^−1^ sodium hydroxide, diluted to 1000 mL with deionized water, and mixed completely. After ultrasound, a buffer solution with pH 6.8 ± 0.1 was obtained. The buffer solution and deionized water were used to dilute 1000 ppm carvone solution into 5 mg L^−1^ carvone aqueous solution and mixed completely. The carvone aqueous solution was placed into a quartz photolysis reaction tube, capped, and placed into the photochemical reaction device, with a reaction temperature of 25 °C. Two light sources were used, a 4.9 A, 500 W mercury lamp and a 5.1 A, 500W xenon lamp, both with a light intensity of 5200 lux. The spectrum of both the mercury lamp and the xenon lamp is shown in Figure 2. As a control, a quartz photolysis reaction tube was wrapped in aluminum foil. The photolysis device was located in a dark, closed space to eliminate the interference of other light sources.

### 2.6. Sample Extraction and Purification

All soil and water samples were extracted and purified using the QuEChERS method [30]. Briefly, a homogenized 5 g sample was placed into a 50 mL polypropylene tube with 10 mL n-hexane and 5 mL ultra-pure water and vortexed for 2 min. Then, 2 g MgSO4 and 1 g NaCl were added. After 1 min of vortexing, the sample was centrifuged at 7000 rpm in a high-speed centrifuge for 5 min. After extraction, 1 mL of the supernatant was transferred to a 2.5 mL polypropylene tube containing 40 mg C18 and 100 mg NaSO_4_ and vortexed for 2 min, then centrifuged at 5000 rpm for 5 min. The supernatant (0.5 mL) was aspirated with a sterile syringe and transferred to the injection bottle through a 0.2 μm syringe filter for GC-MS analysis.

### 2.7. Instrumental Analysis 

All samples were analyzed on an Agilent gas chromatograph (7890B)-mass spectrometer (5977B), which used high-purity helium (>99%) as a carrier gas. The temperatures of the ion source, the quadrupole, and the interface were set at 280 °C, 150 °C, and 270 °C, respectively. The GC-MS conditions were optimized by the qualitative analysis of a 100 mg L^−1^ sample in the full-scan mode. The initial column temperature was set at 90 °C for 5 min. The temperature was then raised to 230 °C at a rate of 10 °C min^−1^ and maintained at the final temperature for 1 min. The split ratio was set to 5:1 by means of split injection; the injection volume was 1 μL, and the column flow rate was 1.5 mL min^−1^. The ionization energy of the electron impact ion source (EI) was 70 eV. A quantitative ion and a pair of qualitative ions with the highest relative abundances were selected for each substance based on its NIST mass spectrum. Selective ions were used in the selected ion monitoring (SIM) mode. The solvent delay was set to 6.5 min.

### 2.8. Method Validation and Data Processing 

The method was validated according to the instructions of SANCO/12682/2019 [31]. A series of compound standard solution concentrations and compound matrix standard solution concentrations of 0.01–5 mg L^−1^ carvone and its degradation products were configured. According to Armbruster and Pry [32], the limit of detection (LOD) is defined as the lowest concentration of a component that can be reliably detected with a given analytical method. Additionally, the limit of quantification (LOQ) is the lowest concentration at which the analyte can not only be reliably detected but at which some predefined goals for bias and imprecision are met. The LOQ of each sample can be calculated at a signal-to-noise ratio of 10 [33]. The accuracy and precision of the method were verified using three different spike levels, and the recovery rate (ratio of treated sample concentration to standard concentration) of each spike level was measured five times. The precision was expressed by relative standard deviation (RSD%), and the accuracy was expressed by the recovery (%) of spiked samples. The linearity of the method was verified by analyzing standard working solution samples and matrix standard working solution samples. The selectivity of the method was verified by analyzing blank and spiked samples from low to high concentrations. The uncertainty of the analytical method is usually caused by random effects and system effects in the experiment, which can be calculated using the covering factor K = 2 at a 95% confidence level (SPSS software). The matrix effect was calculated by Equation (1), the recovery and RSD of the matrix sample were calculated by Equations (2) and (3). Dissipation data are calculated using first-order kinetic Equations (4) and (5):(1)$$ME=\frac{{(S}_{M}{-S}_{S})}{S_{S}}$$
(2)$$R\;=\frac{C_{0}-\;C}{C_{0}}\times 100\%$$
(3)$$RSD=\frac{SD}{R}\times 100\%$$
(4)$$C_{t}=C_{0}e^{-kt}$$
(5)$$T_{0.5}=\frac{ln2}{k}$$
where *ME* is the matrix effect, *S_M_* is the slope of the matrix sample, *S_S_* is the slope of the standard sample, *R* (%) is the calculating spiked recovery, *C* (μg kg^−1^) is the concentration of the matrix sample extract, *C*_0_ (mg L^−1^) is the concentration of the standard working solution, *SD* is the standard deviation of the replicate, *C_t_* is the time-dependent concentrations of soil and water, and *K* is the degradation rate constant. 

The standard mixed working solution of 1000 mg L^−1^ carvone and its degradation products were prepared and diluted step by step into 6 samples of different concentrations for injection test, and the test was repeated three times for each concentration. Based on the detection limit and quantitative limit of dihydrocarvone, the spiked recovery experiments of 0.05, 0.5, and 5 mg kg^−1^ were carried out in blank soil and buffer solutions each with a different pH, and each spiked level was repeated 5 times. 

The degradation products were qualitatively analyzed by GC-MS, and their mass spectra were obtained in full-scan mode. Then, the data were compared with the compounds in NIST database to preliminarily identify the degradation products. Standard compounds were tested by the same operation for secondary validation. 

## 3. Results and Discussion

### 3.1. Sample Extraction and Purification

For its nonpolarity, n-hexane is frequently employed as an essential oil extractant [34]. Acetonitrile is also often used as an extractant for pesticides. Therefore, the two solvents were tested in this study. In general, the recovery of carvone and its degradation products in n-hexane was higher than that in acetonitrile, so we chose n-hexane as the extractant.

PSA, C18, and GCB were tested as purifying agents in this study. Aqueous and buffer solutions did not require additional purification operations. Considering the comprehensive recovery rate, 40 mg C18 and 100 mg Na_2_SO_4_ were selected for this study. 

### 3.2. Validation of the Analytical Method

During the analysis of pesticide residue, the chromatographic signal is suppressed or enhanced due to the matrix interference, which is known as the matrix effect [35]. To reduce the influence of the matrix effect, 0.05–5 mg L^−1^ matrix standard solutions were used to analyze the identified conditions by GC-MS, and a calibration standard curve was fitted using quantitative GC-MS analysis software. The results (Appendix A) showed that carvone, carvone camphor, dihydrocarveol, and dihydrocarvone had good linear correlation coefficients (R^2^ = 0.9979–0.9998). The LOQs of carvone and its degradation products in various matrices were in the ranges of 10–50 μg kg^−1^. When the matrix effect is between −20% and 20%, it can be ignored. When between −50% and 50%, the matrix effect is stronger, and the matrix standard curve must be quantified to eliminate the matrix effect on the results. However, the sample preparation method should be re-optimized when ME > 50% or ME < −50% [36]. All matrices had low matrix effects (−20% < ME < 20%), indicating that it can be ignored (Figure 3). As shown in Appendix A, the average recovery of carvone in soil and buffer solution was between 90.2−98.7% ± 1.8−9.6% RSD. The average recovery of dihydrocarveol and dihydrocarvone in S2 soil was between 81.0−99.4% ± 1.6−4.9% RSD. The average recovery of carvone camphor in buffer solution with pH = 7 was within 95.7–104.2% ± 3.8−8.0% RSD. According to the instruction of SANCO/12682/2019 [31], these data indicate that the method can be used for the accurate determination of carvone and its degradation products in soil and water samples. 

### 3.3. Degradation of Carvone in Different Soil Types

The degradation of carvone in soil follows the first-order kinetic equation. The results of degradation kinetic equation, correlation coefficient (R^2^), and degradation half-life (t_0.5_) are shown in Figure 4. The data showed that there is a strong correlation between the degradation of carvone in soil and soil types (r = 0.803, *p* < 0.05). In aerobic, anaerobic, and sterilization conditions, carvone shows the same degradation law in four kinds of soils. The degradation is the fastest in S1 and the slowest in S4. The degradation half-life of non-sterilized soil is significantly shorter than that of sterilized soil, indicating that the degradation of carvone in soil is mainly caused by microorganisms in soil. Biodegradation in soil accounts for 82.69, 99.25, 99.20, and 99.45% of the total degradation, respectively. We can infer that the degradation of carvone in soil is basically biological; the contribution of abiotic processes such as chemical degradation or volatilization is minimal [37].

The degradation half-lives of carvone in S1, S2, S3 and S4 soils treated by anaerobic treatment are 1.8, 2.2, 2.1, and 3.2 d, respectively. Carvone degraded faster in aerobic conditions, indicating that microorganisms are more suitable for aerobic conditions, which is consistent with the research results of Harder and Probian [38]. Although all soils were cultured in the same conditions, the physical and chemical properties of different soils are different, and the reasons affecting the degradation of carvone in soil were very complex. The degradation of carvone in soil is mainly affected by biological factors. The abundance of microbial communities in different soils is different, which may also be the reason for the different degradation rates of carvone in different soils [39]. In order to better understand the degradation behaviour of carvone in the specific factors, different factors were selected for experiments.

### 3.4. Degradation of Carvone in Different Soil Conditions

Carvone degrades more quickly in untreated soil without sterilization. However, in sterilized soils, carvone degraded faster in soil with organic matter removed. Among the four soils tested, the organic matter content of S2 soil was the highest and that of S1 soil was the lowest. However, in the degradation experiment, the degradation rate of S1 soil was higher than that of S3. In previous experiments, the degradation of carvone was fastest in S1, followed by S3, and S3 was a little faster than S4. These results showed that the content of organic matter is closely related to the degradation of carvone. Organic matter can accelerate the degradation of carvone in soil, but carvone will also be adsorbed by soil. The degradation rate will decrease when the content of organic matter is at a high level. This may be because soil microorganisms need organic matter as a carbon source for normal biological activities [40]. After sterilization, there are no microorganisms in the soil, and carvone will be adsorbed by organic matter, resulting in a slower degradation rate [37]. 

According to Figure 5A, carvone shows longer persistence in acidic soils, which is consistent with the findings of Gámiz, et al. [23]. We found that the half-life of carvone is prolonged by 1.07 d in soils with pH 5, 1.81 d with pH 4, and 4.53 d with pH 3. Considering that carvone is a neutral (non-ionized) compound, and pH has little effect on the adsorption of carvone in soil, we observed a decrease in soil microbial respiration. We suspect that soil pH may be affecting microbial activity rather than acting on carvone directly. According to de Carvalho [41], some microorganisms involved in the degradation of carvone are more suited to neutral or alkaline environments, indicating that acidic conditions may not be suitable for the biodegradation of carvone.

Temperature increases the degradation of carvone, and we found a strong correlation between temperature and K value (r = 0.987, *p* < 0.05) (Figure 5). With the increase of temperature, the K value increased and the degradation half-life of carvone decreased. The degradation half-life of carvone increased by 150% at 10 °C compared with room temperature. When the temperature is 50 °C, the degradation half-life is shortened by 51% compared with that at room temperature. We also observed that soil temperature affects soil respiration, which is significantly reduced at low temperatures (10 °C) and high temperatures (50 °C) and increased at 35 °C. Carvone is a volatile organic compound that can easily escape from soil, leading to lower efficacy and higher application cost. At 10–35 °C, the increase in temperature not only enhanced soil microbial activity but also accelerated the volatilization of carvone in soil, while at 35−50 °C, the volatilization rate of carvone in soil was stronger than the decrease in soil microbial activity affected by temperature. Therefore, minimizing the emissive loss of carvone is important to address economic concerns as well as to maximize weed and pest control efficacy and crop growth [42].

Carvone degrades more quickly in wet soils, and we found a strong correlation between soil water content and compound half-life (r = 0.951, *p* < 0.05). When soil water content increased from 0.1% to 30%, the half-life of carvone decreased from 3.2 to 0.48 d, in accordance with other studies [43]. This may be because as the water content of soil increases, the adsorption capacity of the soil decreases, allowing for the faster degradation of carvone. 

### 3.5. Photolysis of Carvone

The photolysis of carvone in aqueous solution conforms to the first-order kinetic curve (Figure 6). At the same light intensity, the degradation of carvone under the xenon lamp was significantly slower than under the mercury lamp. The photolysis half-life under the xenon lamp was 1.81–1.93 d, while the photolysis half-life under the mercury lamp was 0.76–0.83 d. The radiation spectrum of the mercury lamp is between 200–600 nm and that of xenon lamp is between 200–1000 nm. Carvone strongly absorbs ultraviolet radiation with a wavelength of between 200–240 nm, with absorption maxima at 223 nm and 240 nm [44]. The light emitted by the mercury lamp is mainly ultraviolet light with short wavelength and high energy, while the emission spectrum of the xenon lamp is closer to that of sunlight [45]. Under natural light conditions, the environmental behaviour of carvone would likely be more similar to that under xenon lamp irradiation.

The degradation of carvone in buffer solution was slightly faster than that in deionized water (i.e., the dissipation rate has no obvious change in dark conditions), which indicates that the salt ions in aqueous solution will affect the photolysis of carvone. We noted that in the dark condition, the content of carvone continuously decreased and conforms to the first-order kinetic curve, with a degradation half-life is 3.61 d. Carvone has hydrolytic stability, and no hydrolysate was found when we tested the water sample. Carvone, a volatile organic compound, is unstable in aqueous solution and volatilizes easily. Therefore, the correct half-life of carvone under the mercury lamp is between 0.96-1.16 d, and under the xenon lamp is between 3.61–4.13 d.

### 3.6. Identification and Quantification of Carvone Degradation Products

The retention times of carvone camphor, dihydrocarveol, and dihydrocarvone in SCAN mode were 13.062, 13.822, and 13.908 min, respectively. Their chromatograms, structural formulae, and mass spectra are shown in Figure 7. The degradation products obtained are consistent with those previously reported [46,47].

Dihydrocarveol and dihydrocarvone were detected in non-sterilized S2 soil (Figure 8B). However, these two compounds were rarely detected in anaerobic and sterilized conditions, indicating that dihydrocarvone and dihydrocarveol are mainly bio-transformed by soil microorganisms. The highest concentration of dihydrocarvone was 2.07 mg kg^−1^ at 36 h, which decreased thereafter. The highest concentration of dihydrocarveol was 0.067 mg kg^−1^ at 24 h, which was undetectable after 96 h.

Carvone camphor was detected in aqueous solution after photolysis. While the degradation rate of carvone under the mercury lamp was faster than under the xenon lamp, the conversion rate of photolysis product was higher under the xenon lamp than the mercury lamp (Figure 8A). The average conversion of carvone to carvone camphor was 65.9% under the mercury lamp and 85% under the xenon lamp. 

## 4. Conclusions

In this work, a sensitive and effective method for the determination of carvone and its metabolites was established by GC-MS, and the method was successfully applied to study the degradation of carvone in soil and aqueous solution. The results showed that changes in temperature, pH, oxygen, organic matter, water content, and microbial community affected the degradation of carvone in soil. Overall, high temperature, moderate soil moisture, aerobic conditions, neutral pH, high microbial activity, and low organic material (in the absence of microbial activity) all increased the degradation rate of carvone in soil. Dihydrocarvone and dihydrocarvone, which are the degradation products of carvone, were detected in soils. The conversion rate of dihydrocarveol is very low and that of dihydrocarvone is high, but neither are persistent in soil. The stability of carvone in aqueous solution is not strong and it is easy to volatilize. In aqueous solution, carvone is photoisomerized into carvone camphor under light irradiation. The photolysis rate was faster under the mercury lamp (shorter wavelength), while the conversion rate of photolysis products was higher under the xenon lamp (similar to natural light).

Carvone has shown potential as an insecticide, fungicide, antioxidant, and plant growth regulator and is currently in use as a potato germination inhibitor. When applied to plants, carvone will enter the water, atmosphere, and soil, and undergo chemical changes. Because of its volatility, some carvone will likely be lost in the process of application, the rate of which will be affected by environmental factors such as wind speed, temperature, and so on. The volatilized carvone will undergo photochemical reaction under light irradiation. It appears that carvone is not a persistent compound in either soil or water. However, its degradation products may show higher persistence. Subsequent studies should focus on their potential impact on target and non-target organisms to fully assess the environmental and health risks associated with the use of carvone.

## Figures and Tables

**Figure 1 molecules-27-02415-f001:**
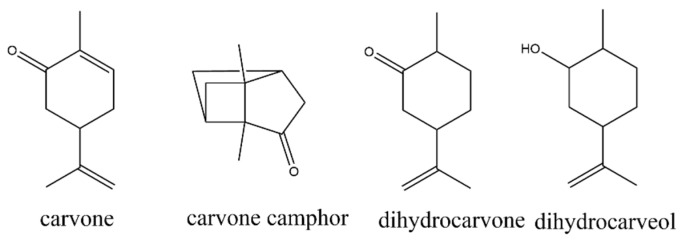
Structures of carvone and its degradation products.

**Figure 2 molecules-27-02415-f002:**
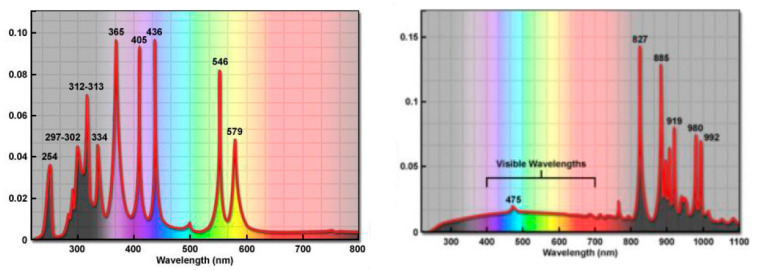
The spectrum of mercury lamp (**left**) and xenon lamp (**right**).

**Figure 3 molecules-27-02415-f003:**
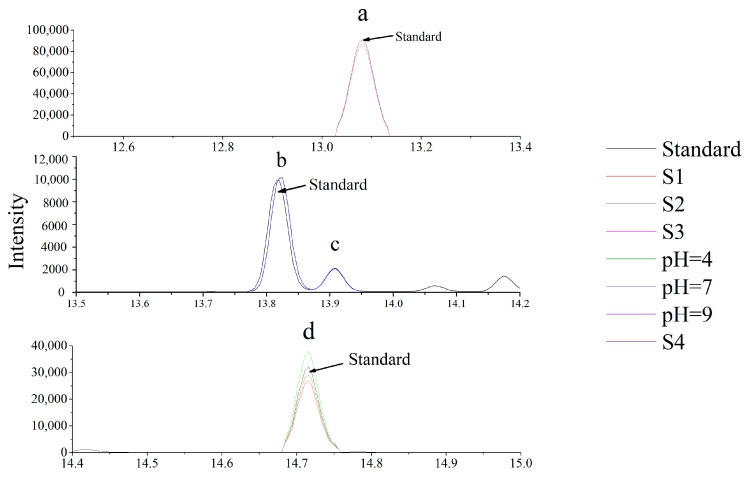
Chromatogram of carvone and its degradation products in the standard solution and matrix solutions at the 5 mg kg^−1^ level. (**a**–**d**) represent the intensities of carvone camphor, dihydrocarveol, dihydrocarvone, and carvone, respectively.

**Figure 4 molecules-27-02415-f004:**
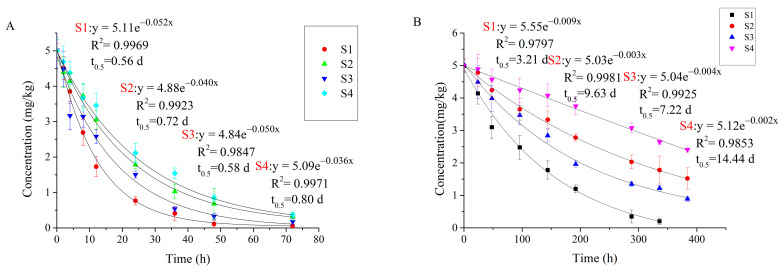
Degradation dynamic curves of carvone in soil under different treatment conditions. (**A**) Degradation under aerobic conditions. (**B**) Degradation under sterilized conditions. (**C**) Degradation under anaerobic conditions.

**Figure 5 molecules-27-02415-f005:**
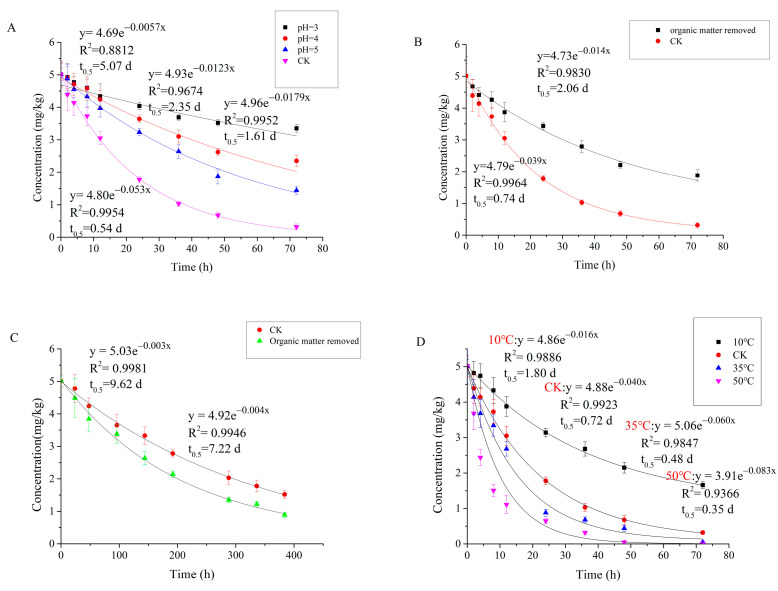
Degradation dynamic curves of carvone in S2 soil under different treatment conditions. (**A**) Degradation under different pH conditions. (**B**) Degradation of organic matter under non-sterilized conditions. (**C**) Degradation of organic matter under anaerobic conditions. (**D**) Degradation under different temperature conditions. (**E**) Degradation under different water content conditions. W1, W2, W3, and W4 represent 0.1, 10, 20, and 30% (W/W) water content, respectively.

**Figure 6 molecules-27-02415-f006:**
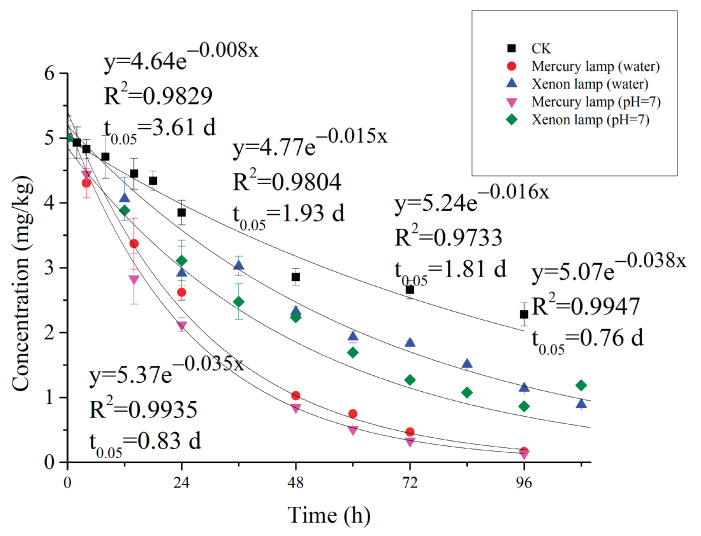
Photolysis dynamics curve of carvone in the different aqueous solutions.

**Figure 7 molecules-27-02415-f007:**
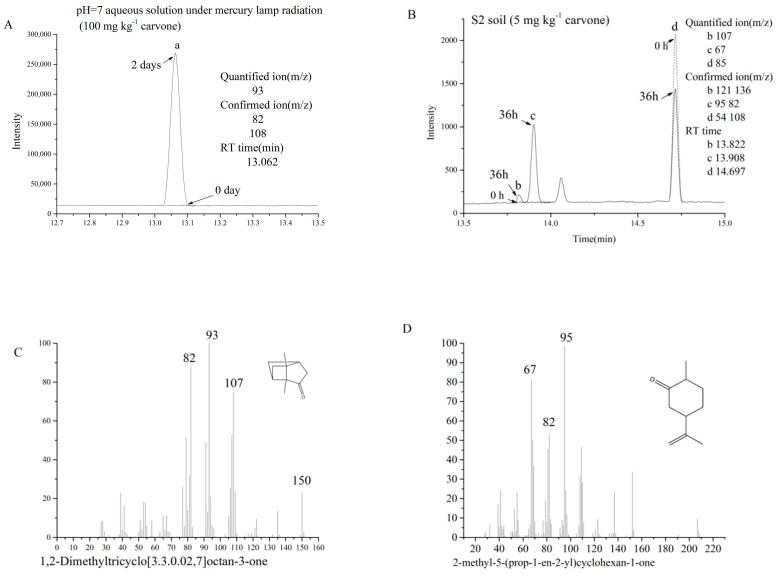
Chromatograms and mass spectra of carvone and its degradation products. (**A**,**B**) Chromatogram of changes in intensity of a, b, c and d, respectively. (**C**–**E**) The standard mass spectra of carvone camphor, dihydrocarvone, and dihydrocarveol, respectively.

**Figure 8 molecules-27-02415-f008:**
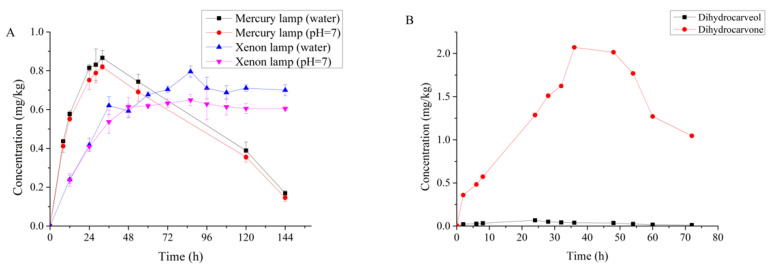
Dynamics of the three degradation products in water (**A**) and soil (**B**). Carvone camphor was detected in aqueous solution, and dihydrocarvone and dihydrocarveol were detected in soil in anaerobic conditions.

**Table 1 molecules-27-02415-t001:** Soil physical and chemical properties.

Soils	Site	Texture				PH	CEC ^a^ (cmol kg^−1^)	OC ^b^ (%)	OM ^c^ (%)
		Sand (%)	Silt (%)	Clay (%)	Texture Class				
S1	Yichang, Hubei (N33°06′, E111°25′)	88.17	7.91	3.92	Sand	7.32	12.1	0.5G	0.86
S2	Chengdu, Sichuan (N30°56′, E105°51′)	51.88	23.35	24.77	sandy clay loam	7.35	25.4	1.0	1.72
S3	Tiangang, Jiling (N43°91′, E126°88′)	17.17	61.66	21.17	Silty loam	6.86	15.7	1.5	2.58
S4	Ningbo, Zhejiang (N29°14′, E121°48′)	42.15	42.83	15.02	loam	7.65	12.9	0.98	1.70

^a^ CEC is the cation exchange capacity; ^b^ OC is the organic carbon content; and ^c^ OM is the organic matter content.

## Data Availability

Not applicable.

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
