# Peer review of "Degradation and Pathways of Carvone in Soil and Water"

_molecules, 2022, doi:10.3390/molecules27082415_

Round 1

Reviewer 1 Report

Why degradation conditions were studied only for S2 soil?

In case of analytical method development for the measurement of carvone and its degradation products, very confusing/unfamiliar terminologies have been used. The proper detail of optimization of QuECHERs should be provided. SIM stands for selective ion monitoring. The method used for LOQ determination is not clear and it should be presented with clarity. Similarly, the methods for calculating spiked recoveries and percentage standard deviations should be presented. Please show the calibration plots for each compound. Moreover, show the chromatograms of the spiked soil before and after degradation.

Reviewer 2 Report

In the present manuscript Huang and co-workers reported a study aiming to understand the fate and degradation of carvone in the environment. The authors described the degradation behavior of carvone in various soil types and in different light conditions in order to quantify the carvone residues and degradation products. Overall, the work is well conceived. The experiments are well described, the methods used are sound and appropriate to the aim of the work. The manuscript provide some interesting data about the environmental impact of carvone and its metabolites. Before accepting the manuscript, however, some minor revisions are needed.
1)    The authors should add in the manuscript a figure reporting the structure of carvone, dihydrocarveol, dihydrocarvone, and carvone camphor; 
2)    Figure 5. The standard mass spectra of dihydrocarveol and dihydrocarvone are not clear. The authors should provide imagine at higher resolution.
3)    Figure 5 caption: the sentence "D-F) The standard mass spectra of carvone, camphor, dihydrocarvone, and dihydrocarveol, respectively" should be "D-F) The standard mass spectra of carvone camphor, dihydrocarvone, and dihydrocarveol, respectively."
4)    Lines 406-407. Please check the sentence “Dihydrocarvone and dihydrocarvone, the degradation products of carvone, were detected in soils.”

Round 2

Reviewer 1 Report

Section 3.8 method validation and data processing, authors should provide a clear description of matrix effect, recovery, LOD, and LOQ.  So far, it relies only on equations and a very confusing description of those equations. The authors should state how the quantities related to each equation were experimentally determined. 

The captions of Figure 6 should also be presented clearly. How figure 6c represents calibration, please provide the values of the standards used. The caption of 6b is also not clearly presented. 
